# Characteristics and Outcomes of COVID-19 Cancer Patients Admitted to a Portuguese Intensive Care Unit: A Case-Control Study

**DOI:** 10.3390/cancers15123264

**Published:** 2023-06-20

**Authors:** Ridhi Ranchor, Nuno Pereira, Ana R. Medeiros, Manuel Magalhães, Aníbal Marinho, António Araújo

**Affiliations:** 1Medical Oncology Department, Centro Hospitalar Universitário do Porto, 4099-001 Porto, Portugal; u10370@chporto.min-saude.pt (M.M.); antonio.araujo@chporto.min-saude.pt (A.A.); 2Internal Medicine Department, Centro Hospitalar Universitário do Porto, 4099-001 Porto, Portugal; u12997@chporto.min-saude.pt; 3Anesthesiology, Intensive Care Medicine and Emergency Department, Centro Hospitalar Universitário do Porto, 4099-001 Porto, Portugal; u13408@chporto.min-saude.pt (A.R.M.); anibalmarinho.sci@chporto.min-saude.pt (A.M.); 4Oncology Research Unit, UMIB—Unit for Multidisciplinary Research in Biomedicine, ICBAS—School of Medicine and Biomedical Sciences, Universidade do Porto, 4050-346 Porto, Portugal

**Keywords:** cancer, COVID-19, intensive care unit, outcomes, mortality

## Abstract

**Simple Summary:**

Cancer patients appear to be more vulnerable to COVID-19. This vulnerability can be explained by an immune-compromised status inherent to oncological disease and antitumoral treatments. Additionally, nosocomial exposure associated with regular medical visits and oncological treatments in the hospital context also contribute to the frailty of this group. Previously published studies have reported a superior mortality in COVID-19 oncological patients as well as more severe clinical evolution, with higher rates of admission to the intensive care unit (ICU) and intubation. However, there are few studies only involving cancer patients with COVID-19 admitted to the ICU. Therefore, this analysis is important from an epidemiological and clinical point of view. Furthermore, the comparison of COVID-19 cancer patients admitted to the ICU with a control group (non-cancer patients) limits the confounding factors and helps to determine whether cancer is a risk factor for adverse outcomes in the intensive care setting.

**Abstract:**

Cancer patients appear to be a vulnerable group in the COVID-19 pandemic. This study aims to compare clinical characteristics and outcomes of cancer and non-cancer patients with COVID-19 admitted to the ICU. All COVID-19 cancer patients (cases) admitted to a Portuguese ICU between March 2020 and January 2021 were included and matched on age, sex and comorbidities with COVID-19 non-cancer patients (controls); 29 cases and 29 controls were enrolled. Initial symptoms were similar between the two groups. Anemia was significantly superior among cases (76% vs. 45%; *p* = 0.031). Invasive mechanical ventilation (IMV) need at ICU admission was significantly higher among cases (48% vs. 7%; odds ratio (OR) = 12.600, 95% CI: 2.517–63.063, *p* = 0.002), but there were no differences for global need for IMV during all-length of ICU stay and mortality rates. In a multivariate model of logistic regression, the risk of IMV need at ICU admission among cases remained statistically significant (adjusted OR = 14.036, 95% CI: 1.337–153.111, *p* = 0.028). Therefore, compared to critical non-cancer patients, critical cancer patients with COVID-19 had an increased risk for IMV need at the moment of ICU admission, however, not for IMV need during all-length of ICU stay or death.

## 1. Introduction

The first cases of pneumonia of unknown origin appeared in Wuhan (China) in December 2019. The etiological agent of this disease, severe acute respiratory syndrome coronavirus 2 (SARS-CoV-2), was identified in January 2020 [1]. Coronavirus disease 2019 (COVID-19) spread rapidly around all world, and in March 2020 the World Health Organization (WHO) declared COVID-19 as a pandemic [2].

The clinical presentation of this disease is variable, from an asymptomatic to a severe and fatal form [3]. The majority of patients are asymptomatic or present mild to moderate symptoms. However, approximately 14% of COVID-19 patients have severe presentations and 5% are critically ill with multiple organ failure and/or death [4,5].

Admission to the intensive care unit (ICU) is crucial for the survival of patients that develop serious forms of disease, especially those who develop respiratory failure (RF) requiring ventilatory support and/or other organ dysfunctions [6]. However, not all critically ill patients are admitted to the ICU. The criteria of admission depend on a patient’s characteristics, disease severity, its reversibility and ICU capacity. Therefore, the number of patients admitted to the ICU differs between hospitals, regions and countries [7,8].

Patients with underlying comorbidities appear to be more vulnerable to COVID-19 [9]. Several comorbidities, such as cardiovascular diseases, chronic kidney disease, liver disease, chronic respiratory diseases, neuromuscular diseases, solid or hematological malignancies and other primary or secondary immunodeficiencies, can influence the three stages of COVID-19 ((1) early viral illness; (2) inflammatory lung injury; (3) post-acute sequelae) by different mechanisms and consequently patients’ evolution and outcomes [10].

In oncological patients this vulnerability can be explained by an immune-compromised status inherent to oncological disease and antitumoral treatments. Additionally, nosocomial exposure associated with regular medical visits and oncological treatments in the hospital context can also contribute to the frailty of this group. Previously published studies have reported a superior mortality among COVID-19 cancer patients admitted to general hospitals as well as more severe clinical evolution, with higher rates of admission to the ICU and intubation [11,12,13,14,15,16,17,18,19,20,21].

The literature published on critically ill COVID-19 patients suggests that cancer patients have an increased risk of death [22]. However, the characteristics and outcomes of COVID-19 cancer patients admitted to the ICU remain largely unknown, with few studies published that focus on this group of patients. Therefore, this study aims to describe the clinical characteristics and outcomes of cancer patients with COVID-19 admitted to the ICU and compare them to non-cancer patients. This analysis is important from an epidemiological and clinical point of view. Furthermore, the comparison of data of critical COVID-19 cancer patients with a control group (patients without cancer) limits the confounding factors and helps to determine whether having an oncological disease is a risk factor for adverse outcomes in the intensive care setting.

## 2. Materials and Methods

### 2.1. Study Design and Patient Selection

A retrospective case–control study was performed on patients with laboratory-confirmed SARS-CoV-2 infection (by the detection of viral RNA in nasopharyngeal swab, using RT-PCR assay) admitted to the ICU of “Centro Hospitalar Universitário do Porto” (CHUPorto) from 2 March 2020 to 31 January 2021.

We included all patients admitted to the ICU in this period that were at least 18 years old and had an oncological disease (solid or hematological) diagnosed prior to hospital admission; patients with skin tumors as well as those with concomitant solid and hematological malignancies were excluded. This group (case group) was matched 1:1 on age, sex and underlying comorbidities with COVID-19 non-cancer patients (control group) admitted to the ICU of CHUPorto in the same period (Figure 1).

### 2.2. Data Collection

Demographic, clinical, laboratorial and radiological findings were retrieved from the electronic health records. Demographic and clinical data included patient’s age, sex, comorbidities, oncological disease’s characteristics (type of cancer, primary site of malignancy, stage of tumor and status of oncological disease), prior oncological therapeutic regimens, source of SARS-CoV-2 infection, symptoms/signs of COVID-19 presentation, organ dysfunctions, COVID-19 pharmacological and non-pharmacological treatments performed in ICU, ventilatory support, complications and length of ICU stay. Laboratorial data collected included the presence of leukopenia, lymphopenia, anemia, thrombocytopenia, elevated C-reactive protein (CRP), elevated procalcitonin (PCT), elevated lactic dehydrogenase (LDH) and elevated D-dimers at ICU admission. Radiologic features on chest-computed tomography (CT) at the moment of ICU admission and laterality of abnormalities were also retrieved. Patients were followed up until they were discharged or died through their electronic health records.

An active oncological disease was defined as histologically confirmed diagnosis of cancer within 5 years prior to the COVID-19 diagnosis or receiving antitumoral treatment (chemotherapy, endocrine therapy, immunotherapy, target therapy, surgery or radiotherapy) within 5 years prior to SARS-CoV-2 infection diagnosis. Cancers were classified according to the 10th revision of the International Classification of Diseases (ICD-10). Septic shock was defined according to the 3rd International Consensus Definitions for Sepsis and Septic Shock.

### 2.3. Statistical Analysis

Statistical analysis was performed using IBM software—SPSS Statistics version 26.0. In a descriptive analysis, continuous variables were presented as mean and standard deviation or median and interquartile range, as appropriate. Categorical variables were presented as absolute number and percentage. The Shapiro–Wilk test was used to test the normality of distributions. The comparison of clinical, laboratorial and radiological findings between the cases and controls was performed by applying parametric (*t*-test) or nonparametric tests (Mann–Whitney test) for quantitative variables, as appropriated, and Chi-squared test for categorical variables. Logistic regression was used to estimate the effect of oncological disease on adverse outcomes (death, IMV need at ICU admission and IMV need during all-length of ICU stay) of COVID-19 patients admitted to the ICU. Odds ratios (ORs) and 95% confidence intervals (CIs) were calculated for study participants with cancer compared to with those without cancer. A *p*-value < 0.05 was considered statistically significant. Initially, the authors performed a univariate analysis followed by a multivariate analysis with variables that exhibited a statistically significant difference on the univariate analysis. The multivariate analysis was adjusted for age, sex and other factors, such as anemia, thrombocytopenia, lymphopenia, elevated procalcitonin (PCT) and elevated D-dimers.

## 3. Results

### 3.1. Demographic and Clinical Characteristics

Twenty-nine COVID-19 cancer patients were hospitalized in the ICU between 2 March 2020 and 31 January 2021. These patients were enrolled in the present study and matched with 29 critical COVID-19 non-cancer patients (controls); 59% (n = 17) of cases were men and the median age was 77 years (63.5–80). Cardiovascular risk factors were the most frequent comorbidities among critical COVID-19 cancer patients. The control group presented a sex distribution, median age and prevalence of comorbidities (variables of matching) very similar to the cases (Table 1).

Of the critical COVID-19 cancer patients, 79.3% (n = 23) had a solid tumor and 20.7% (n = 6) had a hematological cancer. Among solid neoplasms, 13% (n = 3) were at stage IV. According to the ICD-10 classification, malignant neoplasms of breast (13.8%, n = 4), male genital tract (13.8%, n = 4; all of them from prostate) and urinary tract (13.8%, n = 4; three from bladder and one from kidney) were the most common (Table 2). Oncological active disease was present in 69% (n = 20) of participants. A variety of antitumoral treatments were performed prior to ICU admission: 69% (n = 20) received surgery, 27.6% (n = 8) radiotherapy, 27.6% (n = 8) chemotherapy, 20.7% (n = 6) hormone therapy and 3.4% (n = 1) target therapy. In addition, 24.1% (n = 7) and 20.7% (n = 6) of participants received any form of oncological treatment in the 30 days or 14 days preceding COVID-19 diagnosis, respectively.

### 3.2. Clinical Presentation and Laboratorial and Radiological Findings

Patients’ presenting symptoms and laboratorial and radiological findings at the moment of ICU admission are summarized in Table 3. Fever, dyspnea, cough and asthenia were the most common initial symptoms in both groups. There were no significant differences in terms of clinical presentation between the two groups. Lymphopenia and elevated LDH were the most common laboratorial findings in both groups. Anemia was significantly more frequent in cancer patients (n = 22, 75.9% vs. n = 13, 44.8%; *p* = 0.031). Ground-glass opacities were the most common radiological findings and were especially seen non-cancer patients (n = 10, 66.7% vs. n = 16, 100%; *p* = 0.018). There were no statistically meaningful differences between the rates of nosocomial source of SARS-CoV-2 infection (6.9% (n = 2) vs. 3.4% (n = 1), *p* = 0.553).

### 3.3. Treatment and Complications

Organ dysfunctions during ICU stay did not differ between the two groups, as represented in Table 4. All cases and controls had respiratory dysfunction, and cardiovascular dysfunction was the second most common in both groups.

At the moment of admission to ICU, 48.3% (n = 14) of cases vs. 6.9% (n = 2) of controls were on IMV, with a statistically significant difference (*p* = 0.002). However, there was no significant difference when considering the global need for IMV during all-length of ICU stay (75.9%, n = 22 vs. 55.2%, n = 16; *p* = 0.167). The median duration of IMV was 12 (4.75–28.75) vs. 11 (7–14) days, and patients not intubated at admission required IMV after a median time of 2 days in both groups.

COVID-19-related pharmacological and non-pharmacological treatments performed in the ICU are summarized in Table 5. A total of 75.9% (n = 22) of cases received corticosteroids during hospitalization in the ICU vs. 100% (n = 29) of controls (*p* = 0.005). Other drug therapies were used at a similar rate between both groups.

The median length of stay in the ICU was 8 (3–23.50) days for cancer patients and 9 (6–13.50) days for non-cancer patients (p = 0.944). The mortality rate was 58.6% vs. 37.9% for cases and controls, respectively (p = 0.276).

### 3.4. Effect of Oncological Disease on Outcomes

The effect of oncological disease on outcomes (death, IMV need at ICU admission and IMV need during all-length of ICU stay) was analyzed by univariate and multivariate models of logistic regression, as represented in Table 6.

The univariate analysis revealed that critical cancer patients with COVID-19 had a significantly increased risk for IMV need at ICU admission than non-cancer patients (OR = 12.600, 95% CI: 2.517–63.063, *p* = 0.002), but not for IMV need during all-length of ICU stay or death. The other investigated variables did not reach statistically significant differences between groups (Appendix A (Table A1)). In the adjusted multivariate model this significance was maintained (adjusted OR = 14.036, 95% CI: 1.337–153.111, *p* = 0.028).

## 4. Discussion

Cancer patients’ admission to the ICU depends on multiple factors, such as their previous performance status, comorbidities, severity of the condition that motivates ICU admission, as well as its reversibility, and the prognosis associated with the oncological disease. The limited human and material resources seen during the COVID-19 pandemic in addition to the poor prognosis conferred by some oncological diseases may have conditioned the rates of ICU admissions of cancer patients and thus the sample size of our study.

Age, sex and underlying comorbidities were the variables of matching. The median age observed in the cancer group was 77 years. The prevalence of comorbidities tends to increase with aging [22]. Several chronic diseases influence the pathological mechanisms of COVID-19 and consequently patient’s vulnerability to SARS-CoV-2 infection [10,23]. Therefore, not surprisingly, the prevalence of other chronic diseases was high amongst both groups.

The immunosuppression inherent to oncological diseases is variable. Hematological malignancies compromise the immune system, resulting in a higher susceptibility to infections [24,25,26]. In the case of solid tumor immunosuppression results, in the majority of cases, oncological treatments are administered. Studies previously published that aimed to characterize the oncological population with COVID-19 found that lung cancer was one of the most prevalent subtypes [11,12,18,25,26,27,28,29,30,31,32]. In our study, which only included patients admitted to the ICU, the most common tumors were breast cancer, male genital tract cancer and urinary tract cancer. We did not have cases of lung cancer. Patients with lung cancer have an inferior pulmonary reserve, a more severe form of COVID-19-related-lung injury, and thus a lower probability of reversing this condition [33]. These phenomena occur because the main site affected by SARS-CoV-2 is already compromised by lung cancer and/or oncological treatments [32], which, in turn, renders the benefit from invasive measures smaller and contributes to explaining the differences between the prevalence of cancer subtypes in the general population and the oncological patients admitted to the ICU.

Given an increased contact with hospital settings, we were expecting to find a higher prevalence of health-care-associated SARS-CoV-2 infections in the cancer group. However, there was no statistical difference regarding the source of infection between cancer patients and controls. This may be explained by measures adopted at the beginning of the pandemic, such as teleconsultation regimens (especially for patients under surveillance), the concentration of medical appointments/exams/treatments in order to avoid multiple hospital contacts, prolonged prescriptions and the implementation of viral RNA research by RT-PCR assay prior to hospital admissions or treatments.

Lymphopenia is considered a prominent laboratorial marker of severe COVID-19 and was the most common analytical alteration observed in both groups [34,35]. The second most frequent laboratorial abnormality was the LDH elevation. In severe SARS-CoV-2 infection, multiple organ dysfunction leads to cellular hypoxia and cell damage, which, conjugated with lung injury due to acute respiratory distress syndrome (ARDS), culminate with LDH release [36]. We highlight that anemia is not one of the most common findings in SARS-CoV-2 infection, but it occurred in 76% of cancer patients, which was significantly superior to the frequency observed in the control group [37]. This finding suggests that, in addition to the inflammatory component subsequent to a severe infection in a critically ill patient, oncological disease could also have an impact either by direct medullar invasion, medullar suppression following oncological treatments and/or by cancer-associated malnutrition.

According to Liang et al., COVID-19 cancer patients presented a higher risk of severe events (ICU admission and need for IMV and death) than non-cancer patients—39% vs. 8% [12]. However, the authors highlight that the comparison between cancer and non-cancer patients did not consider confounding factors, such as age, sex and comorbidities. In order to minimize the influence of these confounding factors we matched these variables between critical COVID-19 cancer patients and controls.

The mortality rates reported in previously conducted studies ranged from 10% to 84% in the COVID-19 population admitted to the ICU [38]. When considering the oncological population admitted to the ICU it ranged from 25% to 100%, with a pooled mortality of 60.2% in a meta-analysis which included 28 studies (1 276 patients) [39]. In the present analysis, 59% of critical COVID-19 cancer patients died, which is in concordance with the previously mentioned data. According to the available literature, COVID-19 cancer patients admitted to the ICU had a significantly higher mortality compared to non-cancer patients (59.8% vs. 42.3%) [39]. In the present study, the mortality was also superior amongst cancer patients, but did not reach statistical significance (59% vs. 38%, *p* = 0.276). Nadkarni et al. and Plais et al. reported that the presence of an oncologic disease significantly increased the risk of death (OR = 4.144, 95% CI: 1.24–13.83) [39,40]. However, the logistic regression model applied here revealed that the impact of cancer on mortality was not significant. The small sample size could help to explain the different results. The time from ICU admission to death was superior in controls (8 vs. 13 days), suggesting a more rapid evolution to death in the oncological patients’ group, although without statistical significance.

IMV need at the moment of ICU admission was significantly higher among cancer patients (48% vs. 7%, *p* = 0.002). The global need for IMV during all-length of ICU stay was superior in the cancer patients, but without statistical significance (76% vs. 55%, *p* = 0.167). The level of significance persisted in a multivariate logistic regression. These findings suggest that cancer patients evolve more rapidly to severe RF that requires IMV at ICU admission, but not more frequently than those without an oncologic disease.

We expected to find a superior rate of infectious complications among cancer patients, due to the additional immune-compromised status inherent to oncological disease and antitumoral treatments. Nevertheless, the rate of secondary infections did not differ significantly between the two groups (69% vs. 59%, *p* = 0.414). In addition to the immunosuppression conferred by a severe SARS-CoV-2 infection, the high rate of corticosteroid administration in the control group (76% vs. 100%, *p* = 0.005) probably contributed to the susceptibility to secondary infections.

COVID-19 and cancer are both prothrombotic diseases [41]. Therefore, we were expecting to find an increased rate of thromboembolic events in the case group. However, the occurrence of these events was identical (3% vs. 10%, *p* = 0.300). This finding might be associated with the prophylactic usage of an intermediate dose of low-molecular-weight heparin, popular during the pandemic.

The authors remind that the present study has limitations: we conducted a retrospective analysis, with a small sample size, with a 1:1 matching. In addition, to increase sample size, the authors had to include patients with both solid and hematological malignancies, thus providing heterogeneity to the sample, as the courses are different between those types of tumors. In consequence, further prospective, multicentric studies with larger samples and arms differentiating hematological and solid malignancies, if possible, are necessary to reinforce the results obtained here.

## 5. Conclusions

Our findings showed that cancer patients had an increased risk for IMV need at the moment of ICU admission, however, not for IMV need during all-length of ICU stay or death. Nevertheless, the mortality of COVID-19 cancer patients was not superior to that of non-cancer patients, stressing the importance of an aggressive early treatment of oncologic patients. The decision on ICU admission must be individualized, considering the performance status of the patient, the stage of neoplasm, the intention of oncological treatments and the severity and potential of reversibility of COVID-19. Further multicentric studies are needed to shed light on why critical COVID-19 patients with cancer evolve more rapidly to IMV.

## Figures and Tables

**Figure 1 cancers-15-03264-f001:**
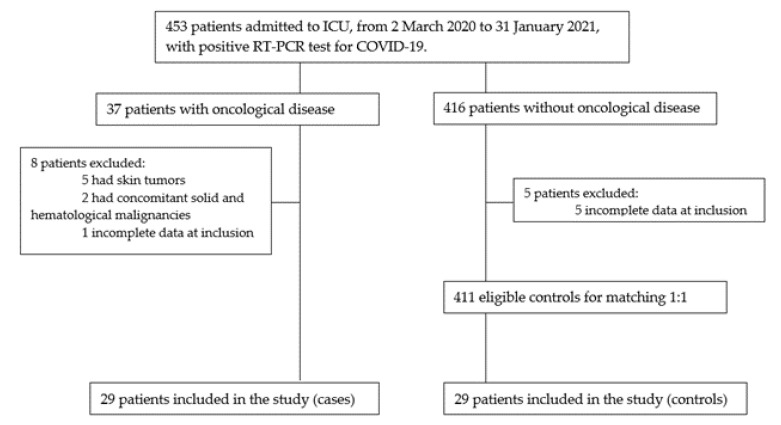
Flowchart of potentially eligible participants in the study.

**Table 1 cancers-15-03264-t001:** Baseline demographic and clinical characteristics of matched case and control groups.

Characteristics of Matching	With Cancer—Case (n = 29) n (%)	Without Cancer—Control (n = 29) n (%)	*p*-Value
**Age (years)** Median (range)	77 (63.5–80)	74 (64.5–80.5)	0.988
**Male Sex**	17 (59%)	17 (59%)	1
**Comorbidities** Current or past smoker Overweight/obesity Arterial hypertension Diabetes mellitus Dyslipidemia Cardiovascular disease Respiratory disease Kidney disease			
	11 (38%) 13 (45%) 19 (66%) 12 (41%) 11 (38%) 9 (31%) 9 (31%) 6 (21%)	10 (34%) 14 (48%) 19 (66%) 14 (48%) 10 (34%) 11 (38%) 10 (34%) 6 (21%)	0.785 0.792 1 0.792 0.785 0.783 0.780 1

**Table 2 cancers-15-03264-t002:** Oncological characteristics of case group.

Patient Characteristics	With Cancer—Case (n = 29) n (%)
**Type of cancer** Solid Hematological	
	23 (79.3%)
	6 (20.7%)
**Stage of solid malignancy** I–III IV	20 (87%)
	3 (13%)
**Cancer classification according to ICD-10** Malignant cancer of digestive tract (C15–C26) Malignant cancer of respiratory and intrathoracic organs (C30–C39) Malignant cancer of bone and articular cartilage (C40–C41) Malignant cancer of soft tissue (C45–C49) Malignant cancer of breast (C50–C50) Malignant cancer of male genital tract (C60–C63) Malignant cancer of the urinary tract (C64–C68) Malignant cancer of central nervous system (C69–C72) Malignant cancer of thyroid and other endocrine glands (C73–C75) Leukemia (C81–C96) Lymphoma (C81–C96) Multiple myeloma (C81–C96)	
	3 (10.3%) 2 (6.9%) 1 (3.4%) 3 (10.3%) 4 (13.8%) 4 (13.8%) 4 (13.8%) 1 (3.4%) 1 (3.4%) 3 (10.3%) 2 (6.9%) 1 (3.4%)
**Status of cancer** Surveillance Active	9 (31%) 20 (69%)
**Prior treatment** Surgery Radiotherapy Chemotherapy Endocrine therapy Target therapy	20 (69%) 8 (27.6%) 8 (27.6%) 6 (20.7%) 1 (3.4%)

**Table 3 cancers-15-03264-t003:** Clinical presentation and laboratorial and radiological features of matched case and control groups.

Patient Characteristics	With Cancer—Case (n = 29) n (%)	Without Cancer—Control (n = 29) n (%)	*p*-Value
**Symptoms or signs** **at initial presentation** Fever Dyspnea Cough Asthenia Dysgeusia Diarrhea Prostration Desaturation Abdominal pain Odynophagia Headache Myalgias			

	13 (44.8%) 12 (41.4%) 10 (34.5%) 5 (17.2%) 3 (10.3%) 3 (10.3%) 2 (6.9%) 2 (6.9%) 1 (3.4%) 1 (3.4%) 1 (3.4%) 1 (3.4%)	13 (44.8%) 16 (55.2%) 18 (62.1%) 6 (20.7%) 2 (6.9%) 2 (6.9%) - 1 (3.4%) 1 (3.4%) 2 (6.9%) 2 (6.9%) 5 (17.2%)	1 0.431 0.650 0.738 0.640 0.640 - 0.553 1 0.553 0.553 0.194
**Analytical parameters at ICU admission** Leukopenia ^a^ Lymphopenia ^b^ Anemia ^c^ Thrombocytopenia ^d^ Elevated CRP ^e^ Elevated PCT ^f^ Elevated LDH ^g^ Elevated D-dimers ^h^			
	5 (17.2%) 24 (82.8%) 22 (75.9%) 5 (17.2%) 19 (65.5%) 13 (44.8%) 25 (86.2%) 14 (48.3%)	2 (6.9%) 25 (86.2%) 13 (44.8%) 4 (13.8%) 19 (65.5%) 10 (34.5%) 24 (82.8%) 15 (51.7%)	0.423 0.717 0.031 0.717 1 0.592 0.717 0.793
**Findings on chest X-ray at ICU admission** Unilateral Bilateral	1 (3.4%) 28 (96.6%)	2 (6.9%) 27 (93.1%)	0.553
**CT pattern at ICU admission** Consolidation Ground-glass opacities Nodular pattern	11/15 (73.3%) 10/15 (66.7%) 1/15 (6.7%)	8/16 (50%) 16/16 (100%) 1/16 (6.3%)	0.273 0.018 0.963
**SAPS II at ICU admission** **SOFA at ICU admission**	44.24 ± 18.27 6 (2–8)	34.31 ± 19.99 4 (2–9)	0.067 0.396

CRP, C-reactive protein; CT, computed tomography; LDH, lactic dehydrogenase; PCT, procalcitonin; SAPS II, simplified acute physiology score; SOFA, sequential organ failure assessment. ^a^ Leukocytes < 4.000 × 10^9^/L; ^b^ lymphocytes < 1.500 × 10^9^/L; ^c^ hemoglobin < 130 g/L for man or <120 g/L for woman; ^d^ platelets < 150.000 × 10^9^/L; ^e^ CRP ≥ 100 mg/L; ^f^ PCT ≥ 0.500 ng/mL; ^g^ LDH ≥ 225 U/L; ^h^ D-dimers ≥ 1000 ng/mL.

**Table 4 cancers-15-03264-t004:** Clinical course during ICU stay of matched case and control groups.

Characteristics	With Cancer—Case (n = 29) n (%)	Without Cancer—Control (n = 29) n (%)	*p*-Value
**Organs/Systems Dysfunction** Respiratory Cardiovascular Liver Renal Hematological CNS	29 (100%) 22 (75.9%) 12 (41.4%) 19 (65.5%) 8 (27.6%) 11 (38%)	29 (100%) 20 (69%) 14 (48.3%) 16 (55.2%) 10 (34.5%) 12 (41.4%)	1 0.770 0.792 0.592 0.777 0.914
**Secondary Infection** VAP CVC-associated infection	20 (69%) 12 (60%) 1 (5%)	17 (58.6%) 9 (52.9%) 1 (5.9%)	0.414
**Septic shock**	17 (58.6%)	14 (48.3%)	0.599
**PTE**	1 (3.4%)	3 (10.3%)	0.300
Length of ICU stay (days)	8 (3–23.50)	9 (6–13.50)	0.944

CNS, central nervous system; CVC, central venous catheter; PTE, pulmonary thromboembolism; UTI, urinary tract infection; VAP, ventilator-associated pneumonia.

**Table 5 cancers-15-03264-t005:** Ventilatory support, therapeutical regimens and clinical outcomes of matched case and control groups.

Variables	With Cancer—Case (n = 29) n (%)	Without Cancer—Control (n = 29) n (%)	*p*-Value
**Oxygen therapy/ventilatory support at ICU admission** Oxygen therapy HCM HFOT NIMV IMV			
	11 (37.9%) 3 (10.3%) 8 (27.6%) 4 (13.8%) 14 (48.3%)	23 (79.3%) 2 (6.9%) 21 (72.4%) 4 (13.8%) 2 (6.9%)	0.186 1 0.002
**IMV need during all-length of ICU stay**	22 (75.9%)	16 (55.2%)	0.167
**Days from ICU admission to IMV** **Total days on IMV**	2 (1–9) 12 (4.75–28.75)	2 (1.75–2) 11 (7–14)	0.685 0.871
**Drug Therapy** Corticosteroids Remdesivir Antimicrobial treatments Vasopressor support	22 (75.9%) 5 (17.2%) 20 (69%) 20 (69%)	29 (100%) 6 (20.7%) 16 (55.2%) 17 (58.6%)	0.005 0.738 0.279 0.414
**Prone positioning**	16 (55.2%)	19 (65.5%)	0.421
**ECMO**	1 (3.4%)	1 (3.4%)	1
**RRT**	2 (6.9%)	1 (3.4%)	0.998
**Days from ICU admission and death**	8 (4–25.5)	13 (7–18)	0.299

ECMO, extracorporeal membrane oxygenation; HCM, high-concentration mask; HFOT, high-flow oxygen therapy; IMV, invasive mechanical ventilation; NIV, non-invasive ventilation; RRT, renal replacement therapy.

**Table 6 cancers-15-03264-t006:** Univariate and multivariate logistic regression.

Univariate Logistic Regression	OR	95% CI	*p*-Value ^a^
Death IMV need at ICU admission IMV need during all-length of ICU stay	2.318	0.809–6.644	0.118
	12.600 2.554	2.517–63.063 0.831–7.842	0.002 0.102
**Multivariate Logistic Regression**	**Adjusted OR ^b^**	**95% CI**	***p*-Value ^a^**
IMV need at ICU admission	14.036	1.337–153.111	0.028

CI, confidence interval; IMV, invasive mechanical ventilation; OR, odds ratio; ^a^ *p*-value < 0.05 was considered statistically significant; ^b^ adjusted for sex, age, anemia, thrombocytopenia, lymphopenia, elevated PCT and elevated D-dimer at ICU admission.

## Data Availability

The datasets generated and/or analyzed during the current study are not publicly available. However, they are available from the corresponding author upon reasonable request.

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
