# Peer review of "Characteristics and Outcomes of COVID-19 Cancer Patients Admitted to a Portuguese Intensive Care Unit: A Case-Control Study"

_cancers, 2023, doi:10.3390/cancers15123264_

Round 1

Reviewer 1 Report

The article titled “Characteristics and Outcomes of COVID-19 Cancer Patients Admitted to a

Portuguese Intensive Care Unit: A Case-Control Study” by Ranchor et. al. compares the

outcome of patients with and without cancers admitted to ICU during Covid-19 pandemic. As

correctly mentioned by the authors, there are multiple published studies on cancer patients

during COVID-19 pandemic but limited number of ICU focused studies. So while the idea is

not technically novel, the findings are clinically important for physicians– i.e. increased risk to

the need of invasive mechanical ventilation for cancer patients as compared to others in the

ICU.

1. Were the IMV requirements due to exacerbation of cancer symptoms or covid-19

related comorbidity, since cancer patients also showed lymphopenia and elevated D-dimers?

2. The sample size of the study is very small for any population study (n < 60). Did the

authors check the power of the study? While there is statistical significance observed

in the findings, low sample size studies with generalization of one subset (by

combining all cancer types) makes the findings questionable. Can the authors include

more patient data from additional hospitals/ICU settings of Portugal so as make the

findings more robust?

3. The whole study is based on a broad division of patients who have cancer during

admission vs. those without it (controls). To do so, the authors have merged patients

of varied cancer types – solid as well as hematological cancers together. Such broad

categorizations are not ideal. It assumes that the disease progression of cancer

patients across different cancer types follow a linear path, which is far from the truth.

However, since this is a retrospective study, it comes with its own set of limitations.

This should be included in the discussion nonetheless.

Minor corrections:

1. Abstract - Line 27: Sentence should start with "This" not these.

2. Line 66 - add comma: In oncological patients, this vulnerability..

Minor corrections:

1. Abstract - Line 27: Sentence should start with "This" not these.

2. Line 66 - add comma: In oncological patients, this vulnerability..

Author Response

Response to Reviewer 1 Comments

 Major Comments

 Point 1. Were the IMV requirements due to exacerbation of cancer symptoms or covid-19 related comorbidity, since cancer patients also showed lymphopenia and elevated D-dimers?

Response 1: The symptoms of cancer progression and COVID-19 are both unspecific and general, and, as such, some confusion can be brought upon our work. The authors would like to emphasize that cancer patients included in the study were deemed stable before admission in an outpatient setting and there was no mention to progression of the disease diagnosed during ICU stay in any of them (on patients’ electronic records and mentioned on imaging studies either directed to that suspicion or by other motives for example deteriorating clinical status).  However, the authors could not distinguish between symptom aetiology since the study is retrospective and based upon electronic clinical records which are dependent on the physician that wrote them.

As depicted by the tables 3 and 4, the rate of lymphopenia and elevated D-dimers were not statistically different between groups and the rate of potential complications from COVID-19 like secondary bacterial infections and thromboembolic events were also not statistically different. Also, those 4 variables did not attain statistical meaning in a bivariate analysis as will be depicted by a new table (table A1, Appendix A).

Thus, it is very likely that the requirement for IVM was largely due to clinical characteristics of both patients and direct impact of COVID-19 to lung tissue and not from exacerbation of patient symptoms (as none of the patients had lung cancer and only 3 had stage IV solid malignancy) or complications from COVID-19 such as those described before.

Point 2. The sample size of the study is very small for any population study (n < 60). Did the authors check the power of the study? While there is statistical significance observed in the findings, low sample size studies with generalization of one subset (by combining all cancer types) makes the findings questionable. Can the authors include more patient data from additional hospitals/ICU settings of Portugal so as make the findings more robust?

AND

Point 3. The whole study is based on a broad division of patients who have cancer during admission vs. those without it (controls). To do so, the authors have merged patients of varied cancer types – solid as well as hematological cancers together. Such broad categorizations are not ideal. It assumes that the disease progression of cancer patients across different cancer types follow a linear path, which is far from the truth. However, since this is a retrospective study, it comes with its own set of limitations. This should be included in the discussion nonetheless.”

Response 2 and 3: Our desired power was 80% for a p value of 0.05. For the lowest OR obtained in the multivariate analysis (~2) and given that the expected proportions of exposure in controls is nearly 0, the sample needed would be 500 which is virtually impossible to achieve in an unicentric retrospective study with cancer patients admitted to an ICU. Furthermore, the generalization between cancer and non-cancer patients, even though questionable was made so we could increase the sample size given that all patients with cancer, at the time the study was done, were being deprived (maybe wrongly) of admissions in ICU given the shortage of resources with multiple patients needed to be admitted in level 2 or higher settings.

While the authors agree that the courses of disease are not linear between solid and hematological malignancies or even within both subtypes, they included all of them due to issues with sample size. Ideally, we would have like to separate patients by solid and hematological malignancies and by primary site of malignancy if possible but that would split our small sample even more thus decreasing the chance of possible meaningful findings. As the authors access this is a major limitation to the study, they made corrections to the last paragraph of the “Discussion” section of the manuscript as suggested by Reviewer 1.

The authors did not engage in any conversations with other hospitals/ICU settings in Portugal to expand this work since that would take much effort with the project having to be approved by multiple ethic commissions (either from the hospitals engaged or by other regulatory committees) which would have made this study inexecutable on deadlines previously agreed between authors. We recognize this fact as being harmful to our study and refer to the need for prospective multicentric studies on the last paragraph of the “Discussion” section.

Minor Comments

Point 4. Abstract - Line 27: Sentence should start with "This" not these.”

Point 5. Line 66 - add comma: In oncological patients, this vulnerability…”

Response 4 and 5: Both corrections were made in the new manuscript available, as suggested.

Reviewer 2 Report

Keywords

Must be 5 words not more. 

INTRODUCTION

1.     The last paragraph of the introduction should be given to the aims of the study.

2.     What is the hypothesis of the study?

3.     What is the scientific significance and originality brought to the field by this study?

MATERIALS AND METHODS

1.     IRB information should be added.

2.     A clear description of the logistic regression is missed. 

RESULTS & DISCUSSION

1.     OR can be added to all the comparison results to present the risk effect of the investigated variables.

2.     Did any adjustment performed on the analysis if so it must be added to the tables.

DISCUSSION 

The discussion must be shorter, and authors must only refer to the literature relevant to their study design and results.

CONCLUSION

It must be improved authors should only refer to their important results.

English can be improved.

Author Response

Response to Reviewer 2 Comments

Point 1. Keywords: Must be 5 words not more.

Response 1: The authors agree with the suggestion. The corrections were made in the new manuscript. Furthermore, the authors would like to explain that they wrote more than 5 words because the template provided by CANCERS allowed it.

Point 2. The last paragraph of the introduction should be given to the aims of the study.

Response 2: The last paragraph of the “Introduction” section already contains the aims of the study which are to “describe the clinical characteristics and outcomes of cancer patients with COVID-19 admitted to the ICU and compare them to non-cancer patients.” Thus, the authors made no alterations on the original manuscript regarding this review.

Point 3. What is the hypothesis of the study?

Response 3: The hypothesis of the study is that critical COVID-19 patients with an oncological disease will have a higher risk for worst outcomes (namely need for IVM, secondary bacterial infections, thromboembolic disease and mortality) than critical COVID-19 patients without an oncological disease. The authors think this fact is well understood within the study design so they made no alterations on the manuscript as they were not deemed mandatory by reviewer 2.

Point 4. What is the scientific significance and originality brought to the field by this study?

Response 4: As mentioned by the authors in the manuscript, there are multiple published studies on cancer patients during the COVID-19 pandemic but limited number of ICU focused studies. Although the idea is not technically novel, the findings are epidemiologically and clinically important for physicians so they can better understand who are the patients who may benefit more from admission on an ICU.

Point 5. IRB information should be added.

Response 5: According to the template provided by CANCERS, IRB information as well as Informed Consent Statement are positioned after the conclusions. To avoid repetition of information the authors did not add this information in the “Material and Methods” section.

Point 6. A clear description of the logistic regression is missed.

Response 6: The authors agree that current bivariate analysis and logistic regression are not well described and they tried to fix this in the new manuscript namely on the last paragraph of the “Material and Methods” section.

Point 7. OR can be added to all the comparison results to present the risk effect of the investigated variables.

Response 7: The authors have the OR of all investigated variables. However, in the bivariate analysis we only presented the OR for the variables of interest (death, IMV need at ICU admission, IMV need during all-length of ICU stay) which is the main objective of the work as a means to avoid the dispersion of readers′ attention and a very extended manuscript. Despite, as more reviewers tend to agree that the divulgence of the ORs for all investigated variables would be beneficial, we opted to include a supplementary table as an appendix (Appendix 1) on the new manuscript. The multivariate analysis was only performed to the for the variables of interest (death, IMV need at ICU admission, IMV need during all-length of ICU stay) that reached statistical significance in the univariate analysis, adjusting for age, sex and other factors mentioned in the label of table 6.

Point 8. Did any adjustment performed on the analysis if so it must be added to the tables.

Response 8: The authors first performed a bivariate analysis. Then performed a multivariate analysis for the variables of interest (death, IMV need at ICU admission, IMV need during all-length of ICU stay) that had statistical significance in the univariate analysis, adjusted for age, sex and other factors mentioned in the label of table 6. We did not perform any other statistical corrections to the multivariate analysis.

Point 9: The discussion must be shorter, and authors must only refer to the literature relevant to their study design and results.

Response 9: The authors agree with the suggestion. The corrections and shortening of the text are available in the new manuscript at the “Discussion” section.

Point 10: It must be improved authors should only refer to their important results.

Response 10: The authors agree with the suggestion. The corrections are available in the new manuscript at the “Conclusions” section.

Point 11: Comments on the Quality of English Language: English can be improved.

Response 11: As English is not the native language of any of the authors, they apologize for the errors. The words and phrase syntax were revised and alterations were performed throughout the text. These are available in the new submitted manuscript.

Reviewer 3 Report

Thank you for giving me the opportunity to review this manuscript. Although the concept idea was interesting, I am of great concern with the presentation of methods and statistics of the paper. I provide reasons for this justification:

1. Epidemiologically, a case-control study is a study that has the cases of the intended main outcome and a control that is free of the disease. In this case, I think that the cases should be of those with COVID-19 and the control as those without COVID-19 admitted to ICU. The patients with cancer and without cancer should be an attribute (independent-variable) or a sub-group of comparison that could contribute to synthesize the odds of the disease risk or outcomes. A sub-analysis within this attribute could then be pulled to understand what are the hard-endpoints of patients with cancer and COVID-19, and those without cancer but with COVID-19.

2. A causal inference is warranted, as those with cancer would definitely have higher risk of hard-endpoints (e.g. deaths) from the ICU. It would be good if authors could first identify the pathways through systematic methods (e.g., directed acyclic graphical methods) before attempting to identify the odds of outcomes of intended variables.

3. Statistically, authors choose to strategize their bivariate analysis results without reporting the odds ratios or effect sizes but only represented them in the multivariate path. The 95% confidence interval was substantially high, showing evidence that the above argument is somewhat correct.

4. Following the third argument above, the sample size could have substantially caused a reduced power of the analysis. While I understand that ICU admissions would be relatively low as compared to other general admissions, authors should have considered a multi-center study.

The English language is fine, some minor editions are required.

Author Response

Response to Reviewer 3 Comments

Point 1: Epidemiologically, a case-control study is a study that has the cases of the intended main outcome and a control that is free of the disease. In this case, I think that the cases should be of those with COVID-19 and the control as those without COVID-19 admitted to ICU. The patients with cancer and without cancer should be an attribute (independent-variable) or a sub-group of comparison that could contribute to synthesize the odds of the disease risk or outcomes. A sub-analysis within this attribute could then be pulled to understand what are the hard-endpoints of patients with cancer and COVID-19, and those without cancer but with COVID-19.

Response 1: The authors understand the argument of reviewer 3 but would like to remind that the main aim of the study was to compare patients with an oncological disease to non-cancer patients. That could have been attained with a subgroup analysis of a larger study but that would hinder the inclusion and exclusion criteria and would have impact on the size of the sample required.

Point 2: A causal inference is warranted, as those with cancer would definitely have higher risk of hard-endpoints (e.g. deaths) from the ICU. It would be good if authors could first identify the pathways through systematic methods (e.g., directed acyclic graphical methods) before attempting to identify the odds of outcomes of intended variables.

Response 2: The authors identified the higher risk that cancer patients have for hard-endpoints from ICU and expected to find a difference between them in this case-control study since the only difference between arms was the presence of an active oncological disease. Nevertheless, that was not the case since on the multivariate analysis (depicted on table 6) there were no statistically significant differences for IMV during all-length ICU stay or mortality. This may have to do with the relatively low statistical power of the study given the small size of the population included. We agree this is the essential limitation to our study that hinders the causal inferences that may arise from other investigations made. However, we would like to emphasize that even with all those limitations, the conclusions of our study might pave the way towards other prospective and larger studies that can shed light on some of their findings namely why cancer patients with COVID-19 have an increased risk for IVM at the moment of admission (or, by other words, have the need for IVM sooner than non-cancer patients).

The authors believe the direct acyclic graphical method might have been useful in drafting the study design but never had the chance to draw one. At the current state, we also believe that doing one now would not help the reader to understand the study design nor alter the results and decided on better explain the flow of data by making it more clear throughout the “Material and Methods” section of the new submitted manuscript.

Point 3. Statistically, authors choose to strategize their bivariate analysis results without reporting the odds ratios or effect sizes but only represented them in the multivariate path. The 95% confidence interval was substantially high, showing evidence that the above argument is somewhat correct.”

Response 3: The authors understand the reviewer’s point of view and, since it was a request from other reviewers as well, they opted to portray the OR of all investigated variables on a supplementary table available in Appendix A (table A1) of the new submitted manuscript. Furthermore, we would like to emphasize that the decision to only depict the OR of the bivariate analysis for the variables of interest (death, IMV need at ICU admission, IMV need during all-length of ICU stay) was a means to avoid the dispersion of readers′ attention and a very extended manuscript.

Point 4: Following the third argument above, the sample size could have substantially caused a reduced power of the analysis. While I understand that ICU admissions would be relatively low as compared to other general admissions, authors should have considered a multi-center study.”

Response 4: As previously answered to other reviewer, our desired power was 80% for a p value of 0.05. For the lowest OR obtained in the multivariate analysis (~2) and given that the expected proportions of exposure in controls is nearly 0, the sample needed would be 500, which is virtually impossible to achieve in an unicentric retrospective study with cancer patients admitted to an ICU. The authors did not engage in any conversations with other hospitals/ICU settings in Portugal to expand this work since that would take much effort with the project having to be approved by multiple ethic commissions (either from the hospitals engaged or by other regulatory committees) which would have made this study inexecutable on deadlines previously agreed between authors. We recognize this fact as being harmful to our study and refer to the need for prospective multicentric studies on the last paragraph of the “Discussion” section.

Point 5: The English language is fine, some minor editions are required.

Response 5: As English is not the native language of any of the authors, they apologize for the errors. The words and phrase syntax were revised and alterations were performed throughout the text. These are available in the new submitted manuscript.

Reviewer 4 Report

Cancer patients are susceptible to COVID-19 because of immune compromised status inherent to oncological disease and anti-tumoral treatments. Authors attempt to analyze clinically and epidemiologically involving cancer patients with COVID-19 admitted to the ICU is appreciable. This study further helps in determining whether cancer is a risk factor for adverse outcomes in intensive care setting.

Major Comments

1.    Please provide the expected flow of potentially eligible participants in the study.

2.    Please include a overview of data collection and follow up in the study

3.    Please include a table summarizing clinical measure during course of illness in the ICU (Organs dysfunction, Secondary Bacterial infections, Drug therapies and Length of ICU stay)

4.    In the methods briefly describe the parameters included and other details for univariate and multivariate analysis performed in your case study.

Minor Comments

1.    Please provide a summary or graphical abstract of the entire research study for easy understanding.

Author Response

Response to Reviewer 4 Comments

Major Comments

Point 1. Please provide the expected flow of potentially eligible participants in the study.

Response 1: The authors agree that de information is needed and so they provided a flow chart in the “Material and Methods” section of the new submitted manuscript (Figure 1).

Point 2. Please include a overview of data collection and follow up in the study.

Response 2: We agree that the manuscript was lacking some data collection and follow-up information and made some corrections and alterations to the manuscript. The authors would like to emphasize that the follow-up period was unique to every patient since it occurrence or discharge dependent.

Point 3: Please include a table summarizing clinical measure during course of illness in the ICU (Organs dysfunction, Secondary Bacterial infections, Drug therapies and Length of ICU stay).”

Response 3: The authors have revised tables 4 and 5 to depict the information suggested by reviewer 4, as they can be seen on the new submitted manuscript.

Point 4: In the methods briefly describe the parameters included and other details for univariate and multivariate analysis performed in your case study.”

Response 4: The authors agree with the suggestion made by reviewer 4. A clear description of the logist regression (bivariate and multivariate analysis) can be seen in the last paragraph of the “Material and Methods” section of the new submitted manuscript.

Minor Comments

Point 5: Please provide a summary or graphical abstract of the entire research study for easy understanding.

Response 5: According to the template provided by CANCERS the initial manuscript submitted already has a summary and abstract. Thus, the authors deemed not necessary or useful to add another summary. Although a graphical representation of the study could present useful, the authors believe it may lack some important information that could only be presented by words in a clear, systematic and concise manner.

Round 2

Reviewer 2 Report

Thank you for revising your manuscript and addressing all of my comments, the manuscript is much better now but English still can be improved.  

Can be improved 

Reviewer 3 Report

Thank you for your justifications and revision.

Acceptable